# INDEPENDENCE-AWARE ADVANTAGE ESTIMATION

## ABSTRACT

Most of existing advantage function estimation methods in reinforcement learning suffer from the problem of high variance, which scales unfavorably with the time horizon. To address this challenge, we propose to identify the independence property between current action and future states in environments, which can be further leveraged to effectively reduce the variance of the advantage estimation. In particular, the recognized independence property can be naturally utilized to construct a novel importance sampling advantage estimator with close-to-zero variance even when the Monte-Carlo return signal yields a large variance. To further remove the risk of the high variance introduced by the new estimator, we combine it with existing Monte-Carlo estimator via a reward decomposition model learned by minimizing the estimation variance. Experiments demonstrate that our method achieves higher sample efficiency compared with existing advantage estimation methods in complex environments.

## 1 INTRODUCTION

Policy gradient method (Sutton et al., 2000) and its variants have demonstrated their success in solving a variety of sequential decision making tasks, such as games (Mnih et al., 2016) and continuous control (Lillicrap et al., 2015). The large variance associated with vanilla policy gradient estimator has prompted a series of previous works to use advantage function estimation, due to its variance-minimized form (Bhatnagar et al., 2008), to get a stable policy gradient estimation (Mnih et al., 2016; Schulman et al., 2015a;b; 2017). For a policy $\pi$ and a state-action pair $(s, a)$, all these works estimate the advantage function $A^\pi(s, a)$ by subtracting an estimate of the value function $V^\pi(s)$ from the estimate of Q-value $Q^\pi(s, a)$. The estimation of $Q^\pi(s, a)$ or $V^\pi(s)$ typically involves a discounted sum of future rewards, which still suffers from the high variance especially when facing the long time horizon.

Meanwhile, in many real-world reinforcement learning applications, we observe that not all future rewards have dependency with the current action. For example, consider a simple multi-round game where at the end of each round of this game, the agent will be assigned a reward, representing whether it wins this round. An episode of the whole game consists of multiple independent rounds. In this example, an action in the current round will not affect the rewards in future rounds, and not all rewards received in future states do contribute to the advantage function of the current action. However, most of existing RL methods (Sutton et al., 2000; Mnih et al., 2013; Schulman et al., 2015b) sum all future rewards to evaluate each action without considering their dependency. By identifying the independence between current action and future states in the environment, we are able to take advantage of such independence to reduce the variance of advantage estimation.

In this paper, we propose Independence-aware Advantage Estimation (IAE), an algorithm that can identify and utilize the independence property between current action and future states. We first introduce a novel advantage estimator that can utilize the independence property by importance sampling. The estimator formalizes a dependency factor $C^\pi$, representing the contribution level of each future reward to advantage function estimation. For those states with no dependency on the current action, there will be a close-to-zero dependency factor $C^\pi$, and the importance sampling estimator can reduce the variance of advantage estimation by ignoring the rewards on these states. For those states with a large dependency factor, the importance sampling estimator will potentially increase variance. In order to take advantage of variance reduction caused by small $C^\pi$ while removing the risk of increased variance by large $C^\pi$, we further combine existing Monte-Carlo estimator with the proposed estimator by decomposing the reward into two estimators and learning the optimal

decomposition by minimizing the corresponding estimation variance. Ideally, when facing states with zero dependency on the current action, our model can learn to distribute all the reward into the importance sampling estimator, where the reward can be ignored; when those states yield extremely large $C^\pi$, our model can learn to distribute part of rewards into the Monte-Carlo estimator to reduce the potential high variance caused by importance sampling. Details of our method are described in Section 3, 4 and 5.

Empirically, we show that our estimated advantage function is closer to ground-truth advantage function $A^\pi$ than existing advantage estimation methods such as Monte-Carlo and Generalized Advantage Estimation (Schulman et al., 2015b). We also test IAE advantage estimation in policy optimization algorithms, showing that our method outperforms other advantage estimation methods in sample efficiency. Results of our experiments are reported in Section 8.

Our contributions can be summarized as follows:

- As far as we know, we are the first to explore and utilize the independence property between current action and future states in environments to improve advantage estimation. The independence property can help us ignore the unnecessary high variance parts in Monte-Carlo estimator which do not contribute to advantage function.
- We propose a practical advantage estimation method to identify and utilize the independence property in environments, which achieves better performance than other advantage estimation methods in both tabular settings and function approximation settings.

## 2 BACKGROUND

### 2.1 NOTATIONS & PROBLEM SETTINGS

We consider a finite-horizon Markov Decision Process defined by $(\mathcal{S}, \mathcal{A}, P, R, \rho_0, \gamma, T)$, where $\mathcal{S}$ is the set of states, $\mathcal{A}$ is the finite set of actions, $P : \mathcal{S} \times \mathcal{A} \times \mathcal{S} \to \mathbb{R}$ denotes the transition probability, $R : \mathcal{S} \times \mathcal{A} \to \mathbb{R}$ denotes the reward function, $\rho_0 : \mathcal{S} \to \mathbb{R}$ denotes the distribution of initial state $S_0$, $\gamma \in (0, 1]$ is the discount factor, $T$ is the total time steps. We denote $S_t$, $A_t$, $R_t$ as the random variable of state, action, reward at time $t$, and $\tau_t := (S_t, A_t, R_t, S_{t+1}, ..., S_T, A_T, R_T)$ as trajectory starting from time $t$.

We denote $\pi : \mathcal{S} \times \mathcal{A} \to \mathbb{R}$ as a stochastic policy, and use the notation of $Q^\pi(s_t, a_t)$, $V^\pi(s_t)$, $A^\pi(s_t, a_t)$ as state-action value function, state value function and advantage function respectively. In the following discussions, we will recognize $(s_t, a_t)$ as a constant state-action pair whose advantage function needed to be estimated.

### 2.2 ADVANTAGE FUNCTION ESTIMATORS

Monte-Carlo estimator $\hat{A}_t^{\text{MC}}$ of advantage function $A^\pi(s_t, a_t)$ is formalized below:

$$\hat{A}_t^{\text{MC}} := -V_\theta(s_t) + \sum_{k=0}^{T-t} \gamma^k R_{t+k}, \text{where } \tau_t \sim P^\pi(\tau_t | s_t, a_t).$$

Here $V_\theta(s_t)$ denotes the function approximator of value function $V^\pi(s_t)$. We use $\tau_t \sim P^\pi(\tau_t | s_t, a_t)$ to denote that trajectory $\tau_t$ is generated by policy $\pi$ from $s_t, a_t$.

Some previous work focuses on reducing the variance of $\hat{A}_t^{\text{MC}}$ at the cost of introducing bias (Schulman et al., 2015b), by using the n-step TD estimator and GAE estimator of advantage function $A^\pi(s_t, a_t)$:

$$\hat{A}_t^{\text{TD}(n)} := \begin{cases} -V_\theta(s_t) + \sum_{k=0}^{n-1} \gamma^k R_{t+k} + \gamma^n V_\theta(S_{t+n}), & \text{if } n < T - t \\ \hat{A}_t^{\text{MC}}, & \text{if } n \geq T - t \end{cases}$$

$$\hat{A}_t^{\text{GAE}} := (1 - \lambda) \sum_{n=0}^{\infty} \lambda^n \hat{A}_t^{\text{TD}(n+1)}, \text{where } \tau_t \sim P^\pi(\tau_t | s_t, a_t).$$

## 3    UTILIZING INDEPENDENCE PROPERTY IN ADVANTAGE ESTIMATION

In many cases, we can utilize the independence between current action and future states to avoid unnecessary parts of variance in the Monte-Carlo estimators. Consider the example where we have a current state $s_t$ whose advantage functions with respect to all actions are needed to be estimated. For a set of $s_{t+k}$ which can be reached from $s_t$, we have independence property such that the probability $P^\pi(s_{t+k}|s_t, a_t)$ is a constant with respect to $a_t$. Although the Monte-Carlo return estimator from $s_{t+k}$ may have large variance, it is clear that the return after reaching $s_{t+k}$ gives no contribution to $A^\pi(s_t, a_t)$ in this case.

In this section, we propose a new advantage estimator based on importance sampling, which removes the variance in Monte-Carlo return estimator after $s_{t+k}$ by utilizing independence property, exactly as we described above. In later discussions, we will name the proposed estimator as importance sampling advantage estimator.

By importance sampling approach, we present our way to derive $A^\pi(s_t, a_t)$ into a form which utilizes independence property:

$$A^\pi(s_t, a_t) = \mathbb{E}_{\tau_t \sim P^\pi(\tau_t|s_t, a_t)} \left[ \sum_{k=0}^{T-t} \gamma^k R_{t+k} \right] - \mathbb{E}_{\tau_t \sim P^\pi(\tau_t|s_t)} \left[ \sum_{k=0}^{T-t} \gamma^k R_{t+k} \right]$$

$$= \mathbb{E}_{\tau_t \sim P^\pi(\tau_t|s_t)} \left[ \sum_{k=0}^{T-t} \gamma^k R_{t+k} \left( \frac{P^\pi(S_{t+k}, A_{t+k}|s_t, a_t)}{P^\pi(S_{t+k}, A_{t+k}|s_t)} - 1 \right) \right]. \tag{1}$$

To briefly summarize our derivation, we perform importance sampling in every future time $t + k$, estimating the discounted reward $\gamma^k R_{t+k}$ in distribution $P^\pi(S_{t+k}, A_{t+k}|s_t, a_t)$ by sampling on distribution $P^\pi(S_{t+k}, A_{t+k}|s_t)$. It is worth noting that when $k \geq 1$, $A_{t+k}$ is independently sampled by $S_{t+k}$, and we are able to omit $A_{t+k}$ in the probability ratio.

For the simplicity of discussion, we will use the following definition:

$$C_k^\pi(s_t, a_t, s_{t+k}, a_{t+k}) := \frac{P^\pi(s_{t+k}, a_{t+k}|s_t, a_t)}{P^\pi(s_{t+k}, a_{t+k}|s_t)} - 1. \tag{2}$$

where we call $C_k^\pi(s_t, a_t, s_{t+k}, a_{t+k})$ the dependency factor, since the value captures how taking a specific action $a_t$ changes the probability of reaching a future state-action pair $(s_{t+k}, a_{t+k})$. It is clear from equation 1 and equation 2 that future state pair $s_{t+k}$ that has dependency factor $C_k^\pi(s_t, a_t, s_{t+k}, a_{t+k})$ close to 0 has small contribution to $A^\pi(s_t, a_t)$, which further demonstrates that the rewards from independent future states do not contribute to advantage estimation, even if Monte-Carlo return signal has high variance. In practice, we face the challenge to estimate the dependency factor $C^\pi$ by data samples. We propose a novel modeling method and a temporal difference training strategy to solve this problem, which is detailed in Section 5.

## 4    OPTIMAL COMBINATION WITH MONTE-CARLO ESTIMATOR

The advantage estimation method proposed in section 3 nicely deal with those future rewards which are independent on current action, since we have dependency factor close to zero and further ignore those rewards. However, the importance sampling advantage estimator may badly deal with those rewards with large dependency factor, which can increase the variance in estimation. To illustrate, consider we have $s_t$ whose advantage functions with respect to all actions are needed to be estimated. The Monte-Carlo return starting from $s_t$ following $\pi$ is close to a constant $q$, while there is a large gap between $P^\pi(S_{t+k}|s_t, a_t)$ and $P^\pi(S_{t+k}|s_t)$. This dependent case can cause high variance in importance sampling advantage estimator, even when Monte-Carlo estimation has low variance.

To deal with the potential high variance problem, we seek to find the optimal combination between the proposed importance sampling estimator and Monte-Carlo estimator. There have been some previous works (Grathwohl et al., 2017) focusing on combining two estimators by optimizing a control variate, producing an estimator with less variance. Inspired by that, we decompose the reward into two estimators with a reward decomposition model, and learn the reward decomposition model by minimizing estimation variance.

The following theorem demonstrates our derivation to combine two estimators:

**Theorem 1.** *Suppose $r_{t+k} \sim \hat{R}(r_{t+k}|s_t, a_t, \tau_{t+k})$, where $\hat{R}$ is any probability distribution. Then*

$$A^\pi(s_t, a_t) = \mathbb{E}_{\tau_t \sim P^\pi(\tau_t|s_t,a_t)} \left[ \sum_{k=0}^{T-t} \gamma^k (R_{t+k} - r_{t+k}) \right] - \mathbb{E}_{\tau_t \sim P^\pi(\tau_t|s_t)} \left[ \sum_{k=0}^{T-t} \gamma^k (R_{t+k} - r_{t+k}) \right]$$

$$+ \mathbb{E}_{\tau_t \sim P^\pi(\tau_t|s_t)} \left[ \sum_{k=0}^{T-t} \gamma^k r_{t+k} C_k^\pi(s_t, a_t, S_{t+k}, A_{t+k}) \right] \tag{3}$$

The proof of theorem 1 is provided in Appendix A.1. The sum of all the terms containing $r_{t+k}$ in equation 3 constructs a control variate with zero expectations, while the expectation of rest of terms containing $R_{t+k}$ is the value of advantage function $A^\pi(s_t, a_t)$. From equation 3, we find that $r_{t+k}$ determines the way in which rewards are divided into two estimators. If $r_{t+k}$ is close to 0, rewards are divided into the Monte-Carlo estimator; if $r_{t+k}$ is close to $R_{t+k}$, rewards are divided into the importance sampling advantage estimator. We parameterize $r_{t+k}$ as $r_{t+k,\psi}$, seeking to optimize $\psi$ to gain an advantage estimator with minimized variance. In practice, we represent $r_{t+k,\psi}$ by a neural network $r_\psi(s_t, a_t, S_{t+k}, A_{t+k}, R_{t+k}, k)$, whose architecture is shown by Figure 6 in the appendix.

For simplicity, we will use $\hat{J}_\psi(\tau_t)$ and $\hat{I}_\psi(\tau_t, a_t)$ to denote two random variables inside of expectation in equation 3, which is written by:

$$\hat{J}_\psi(\tau_t) := \sum_{k=0}^{T-t} \gamma^k (R_{t+k} - r_{t+k}); \quad \hat{I}_\psi(\tau_t, a_t) := \sum_{k=0}^{T-t} \gamma^k r_{t+k} C_k^\pi(s_t, a_t, S_{t+k}, A_{t+k}) \tag{4}$$

We use $\mathrm{Var}_\psi(s_t, a_t)$ to denote the variance of advantage estimator derived from equation 3. Though we are not able to write the closed-form of variance in advantage estimation since we don't know how three sampling processes in equation 3 are correlated, we are able to derive the variance upper bound of possible estimators by equation 5, whose proof is shown in Appendix A.2.

$$\mathrm{Var}_\psi(s_t, a_t) \leq 3 \left[ \mathrm{Var}_{\tau_t \sim P^\pi(\tau_t|s_t,a_t)} \left[ \hat{J}_\psi(\tau_t) \right] + \mathrm{Var}_{\tau_t \sim P^\pi(\tau_t|s_t)} \left[ \hat{J}_\psi(\tau_t) \right] + \mathrm{Var}_{\tau_t \sim P^\pi(\tau_t|s_t)} \left[ \hat{I}_\psi(\tau_t, a_t) \right] \right] \tag{5}$$

Based on equation 5, we further derive the upper bound of variance which is friendly to optimize:

$$\mathbb{E}_{a_t \sim \pi(a_t|s_t)} \mathrm{Var}_\psi(s_t, a_t) \leq 6 \mathrm{Var}_{\tau_t \sim P^\pi(\tau_t|s_t)} \left[ \hat{J}_\psi(\tau_t) \right] + 3 \mathbb{E}_{a_t \sim \pi(a_t|s_t)} \mathrm{Var}_{\tau_t \sim P^\pi(\tau_t|s_t)} \left[ \hat{I}_\psi(\tau_t, a_t) \right] \tag{6}$$

We seek to optimize $\psi$ to minimize the variance upper bound shown in equation 6. To achieve this, we firstly use two value function approximators $V_{w_1}(s_t)$, $I_{w_2}(s_t, a_t)$ to approximate the expectation of two estimators respectively:

$$V_{w_1}(s_t) \approx \mathbb{E}_{\tau_t \sim P^\pi(\tau_t|s_t)} \left[ \hat{J}_\psi(\tau_t) \right]; \quad I_{w_2}(s_t, a_t) \approx \mathbb{E}_{\tau_t \sim P^\pi(\tau_t|s_t)} \left[ \hat{I}_\psi(\tau_t, a_t) \right] \tag{7}$$

To approximate the expectation, we use SGD to minimize the mean squared error. The parameter update process can be finally expressed as follows, where $\alpha_w$ represents the step-size:

$$w_1' = w_1 + \alpha_w (\hat{J}_\psi(\tau_t) - V_{w_1}(s_t)) \nabla_{w_1} V_{w_1}(s_t), \tau_t \sim P^\pi(\tau_t|s_t) \tag{8}$$

$$w_2' = w_2 + \alpha_w (\hat{I}_\psi(\tau_t, a_t) - I_{w_2}(s_t, a_t)) \nabla_{w_2} I_{w_2}(s_t, a_t), \tau_t \sim P^\pi(\tau_t|s_t) \tag{9}$$

If we assume that $V_{w_1}(s_t)$ and $I_{w_2}(s_t, a_t)$ accurately represent the expectation, then we get the gradient to optimize $\psi$ as follows, in order to minimize the upper bound of $\mathbb{E}_{a_t \sim \pi(a_t|s_t)} \mathrm{Var}_\psi(s_t, a_t)$:

$$\nabla_\psi \left( 6 \mathrm{Var}_{\tau_t \sim P^\pi(\tau_t|s_t)} \left[ \hat{J}_\psi(\tau_t) \right] + 3 \mathbb{E}_{a_t \sim \pi(a_t|s_t)} \mathrm{Var}_{\tau_t \sim P^\pi(\tau_t|s_t)} \left[ \hat{I}_\psi(\tau_t, a_t) \right] \right)$$

$$= 3 \mathbb{E}_{\tau_t \sim P^\pi(\tau_t|s_t)} \left[ 2 \nabla_\psi (\hat{J}_\psi(\tau_t) - V_{w_1}(s_t))^2 + \sum_{a_t} \pi(a_t|s_t) \nabla_\psi (\hat{I}_\psi(\tau_t, a_t) - I_{w_2}(s_t, a_t))^2 \right] \tag{10}$$

We can use SGD to optimize $\psi$ expressed by equation 11, where $\alpha_\psi$ represents the step-size:

$$\psi' = \psi + \alpha_\psi \Big[ 2(\hat{J}_\psi(\tau_t) - V_{w_1}(s_t)) \nabla_\psi \hat{J}_\psi(\tau_t)$$

$$+ \sum_{a_t} \pi(a_t|s_t)(\hat{I}_\psi(\tau_t, a_t) - I_{w_2}(s_t, a_t)) \nabla_\psi \hat{I}_\psi(\tau_t, a_t) \Big], \tau_t \sim P^\pi(\tau_t|s_t) \tag{11}$$

Here we will illustrate why performing gradient descent on $\psi$ leads to an advantage estimation with lower variance. For a single $r_{t+k,\psi}$, the gradient component in the first term of equation 11 pushes the $r_{t+k,\psi}$ to change towards the direction to the mean total return; meanwhile, the second term of equation 11 counteracts the gradient in the first term, preventing $r_{t+k,\psi}$ from constructing a constant return by the restriction in variance of importance sampling estimator $\sum_{k=0}^{T-t} \gamma^k r_{t+k,\psi} C_k^\pi(s_t, a_t, S_{t+k})$. When we have the independence property in environments (i.e. the value of $C_k^\pi$ is close to zero), the counteraction effect in the second term will disappear. With the gradient in the first term, $r_{t+k,\psi}$ will be rapidly optimized towards the mean total return, making the variance of advantage estimator to dramatically decrease along the training process.

By replacing the expectation terms in equation 3 by function approximators, the form of independence-aware advantage estimator is given by:

$$\hat{A}_t^{\text{IAE}} := \hat{J}_\psi(\tau_t) - V_{w_1}(s_t) + I_{w_2}(s_t, a_t), \text{where } \tau_t \sim P^\pi(\tau_t|s_t, a_t). \tag{12}$$

## 5 Dependency Factor Estimation

The final challenge in our method is to estimate the dependency factor $C^\pi$, which is crucial to make the advantage estimator low-biased. In this section, we will introduce our modeling and training method, which is able to give accurate dependency factor estimation in experiments.

It is hard to estimate the transition probability in equation 2 because of the high dimensionality of state space. Here we derive the ratio between two transition probability into a form which can be represented by an action classifier by equation 13, whose proof is shown in Appendix A.3.

$$\frac{P^\pi(s_{t+k}, a_{t+k}|s_t, a_t)}{P^\pi(s_{t+k}, a_{t+k}|s_t)} = \frac{P^\pi(a_t|s_t, s_{t+k}, a_{t+k})}{\pi(a_t|s_t)} = \frac{P^\pi(a_t|s_t, s_{t+k})}{\pi(a_t|s_t)}, \text{if } k \geq 1. \tag{13}$$

In practice, we use an action classification model $P_\phi(a_t|s_t, s_{t+k}, k)$ to represent $P^\pi(a_t|s_t, s_{t+k})$, using $C_\phi(s_t, a_t, s_{t+k}) := \frac{P_\phi(a_t|s_t, s_{t+k}, k)}{\pi(a_t|s_t)}$ to give an approximation of $C^\pi$. We call $P_\phi(a_t|s_t, s_{t+k}, k)$ dependency model in later discussions.

Similar to the derivation in previous work (Liu et al., 2018), we show that the dependency model can be learned in an temporal difference manner:

$$P^\pi(a_t|s_t, s_{t+k_2}) = \mathbb{E}_{s_{t+k_1} \sim P^\pi(s_{t+k_1}|s_{t+k_2}, s_t)}\left[P^\pi(a_t|s_t, s_{t+k_1})\right], \text{if } k_2 > k_1 \geq 1. \tag{14}$$

Detailed proof of equation 14 is provided in Appendix A.4. Although the case is different from the common cases in temporal difference learning, it is clear that the samples of distribution $P^\pi(s_{t+k_1}|s_{t+k_2}, s_t)$ can be collected by directly rolling out policy $\pi$, then we are able to train model $P_\phi$ by minimizing temporal difference error. We use a mixture of temporal difference target and Monte-Carlo target to train model $C_\phi$, which is detailed in Appendix B. Empirically, we demonstrate the effectiveness of our approach to accurately estimate dependency factor in section 8.2.

## 6 Algorithm

In this section, we present crucial algorithms details of the method presented in Section 3, 4 and 5.

### 6.1 Model Dependency and Pseudo-Code

We illustrate the general framework of our algorithm shown by Figure 1. The model of dependency factor $C^\pi$, lying in the basic part of our algorithm, can be directly trained by samples of current policy $\pi$. Two value functions $I_{w_1}$ and $V_{w_2}$ are trained by SGD on MSE loss, shown in equation 8 and equation 9, given the the reward decomposition $r_\psi$ as input. The reward decomposition model $r_\psi$ is also trained by SGD shown in equation 11, given the output of $I_{w_1}$ and $V_{w_2}$. The pseudo-code of our algorithm is provided by Algorithm 1.

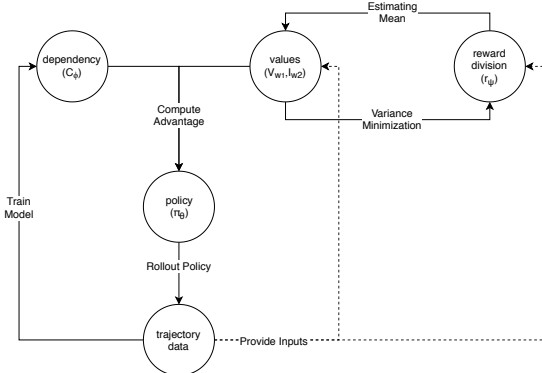

Figure 1: Model Dependency Graph.

---

**Algorithm 1** Policy Optimization Algorithm by Independence-aware Advantage Estimation

---

1: Initialize policy parameters $\theta$, dependency model parameters $\phi$, decomposition model parameters $\psi$, and value function parameters $w_1$, $w_2$.
2: **for** each iteration **do**
3:     Sample trajectory $\tau$ with policy $\pi_\theta$
4:     Update $\phi$ by temporal difference learning
5:     Update $w_1$, $w_2$ to minimize mean squared error by equation 8 and equation 9
6:     Update $\psi$ to minimize variance upper bound by equation 11
7:     Compute advantage estimation $\hat{A}_t^{\mathrm{IAE}}$ at each time-step $t$ by equation 12
8:     Update $\theta$ by Proximal Policy Optimization
9: **end for**

---

## 6.2 COMPUTATIONAL COMPLEXITY

There might be concerns about computational complexity for training dependency model and reward decomposition model, since the training requires $O(T^2)$ number of $(s_t, s_{t+k})$ pairs as training samples. We apply two techniques, trajectory truncation and block-wise training, enabling our algorithm to have a comparable computational complexity with PPO algorithm.

**Trajectory Truncation.** We truncate the trajectory into slices of 128 timesteps similar to the approach in PPO algorithm. We apply our method into the truncated trajectories by considering a reward $Q_\theta(s_t, a_t)$ at the last step. The reward decomposition model also considers the decomposition over the final reward $Q_\theta(s_t, a_t)$.

**Block-wise Training.** When training dependency model and reward decomposition model, we provide a total of $128^2$ numbers of $(s_t, s_{t+k})$ pairs, and accumulate gradient for every valid $(s_t, s_{t+k})$ pair. Only $128 \times 2$ times of forward and backward passes in CNN feature extractor are computed though we deal with a squared number of training data, making our method computationally efficient. Our method requires less than 3 times training time compared to PPO algorithm.

## 7 RELATED WORK

Policy gradient (Sutton et al., 2000) provides the basic form to optimize a parameterized policy in expected returns. Generalized Advantage Estimation (Kimura et al., 2000; Schulman et al., 2015b) replaces Monte-Carlo estimator by the mixture of N-step temporal difference estimator, reducing the variance of policy gradient estimator while introducing bias. Another series of previous works focus on how to optimize parameterized policy without focusing on the selection of estimator. Among these works, TRPO (Schulman et al., 2015a) and PPO (Schulman et al., 2017) are part of the recent works that reaches state-of-the-art performance on a variety of tasks.

Some of previous works have shown that RL algorithm tends to be unstable when planning horizon is long (Jiang et al., 2015). Another series of works (François-Lavet et al., 2015; Xu et al., 2018) focus on how to select discount factor $\gamma$ to improve the performance of RL algorithm. Our method suggests another solution to the problem in RL, since IAE leads to substantial variance reduction in return signal even if the planning horizon is long, which improves the performance while keeping the long planning horizon.

Our method performs gradient descent on estimation variance to improve the estimator as training proceeds. This method has been used in recent works on various applications. One previous work (Hanna et al., 2017) focuses on optimizing a behaviour policy to minimize the variance of off-policy value estimation; another previous work (Grathwohl et al., 2017) focuses on getting the optimal variance balance between REINFORCE estimator and reparameterized gradient estimator by minimizing estimation variance.

In the problem of density ratio estimation, our method is similar to one previous work (Liu et al., 2018) which transforms the density ratio estimation problem into an temporal difference learning problem. Our method is different that we focus on estimating the dependency between current actions and future states in a fixed policy, while this previous work focuses on estimating the ratio of the future state reaching probability between two different policies.

## 8 EXPERIMENTS

In our experiments, we provide empirical results to answer the following questions:

- Can our dependency model training method in section 5 precisely estimates the dependency factor $C^\pi$, and captures the independence property in environments?
- Can our method utilize the independence property to reduce the variance in advantage estimation, and further give more accurate advantage estimation than other advantage estimation methods?
- Can IAE improve the overall performance of policy optimization algorithm, for instance, PPO algorithm?

To answer the first question, we train the dependency model by method in section 5 and compare the prediction with ground-truth value, proving the capacity of our training method to model dependency factor $C^\pi$. This part of results are detailed in section 8.2.

For the second question, we show that IAE method gives advantage estimation with less variance in tabular settings, and reduces value function training error in function approximation settings. In the Pixel Grid World environment, we further show that our method gives advantage estimation closer to ground-truth advantage function than MC and GAE method under cosine similarity metric. This part of results are detailed in section 8.3. To demonstrate how IAE method utilize independence property, we also provide case studies in Appendix C.3 to visualize the effect of variance reduction by low dependency factor in Pixel Grid World environment.

For the last question, we provide training curves in section 8.4 in Pixel Grid Worlds. Compared with PPO algorithm with Monte-Carlo advantage estimation and generalized advantage estimation, IAE method makes policy optimization process more sample-efficient.

### 8.1 ENVIRONMENT SETTINGS

We perform experiment on two types of environments: finite-state MDPs and Pixel Grid World.

**Finite-state MDP settings.** To evaluate the quality of advantage estimation of our method in tabular cases, we construct different 3-state MDPs with different transition probability and reward functions. We categorize state transition probability settings into connected settings and isolated settings, and categorize reward settings into high-variance settings and low-variance settings. Detailed descriptions of transition probability and reward settings are provided in Appendix C.1.

**Pixel Grid World Environment.** To evaluate our method in function approximation settings, we build Pixel Grid World environment where observations are provided by high-dimensional images.

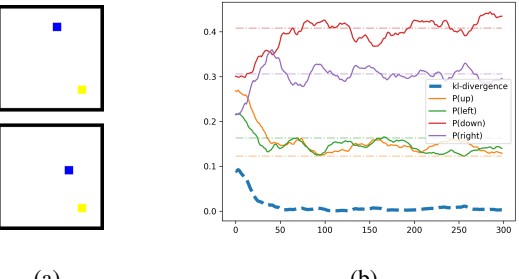 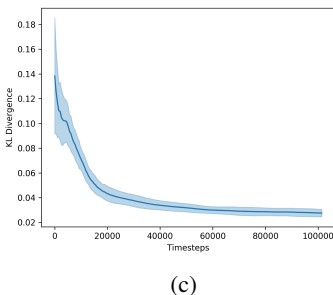

|  (a)  |  (b)  |  (c)  |

Figure 2: Results on dependency factor modeling. (a): The top and bottom image respectively illustrate $s_t$ and $s_{t+k}$ in the case study, and here we set $k$ to be 7. (b): Dashed lines show the true distribution of $P^\pi(a_t|s_t, s_{t+k})$; solid lines show the dependency model prediction $P_\phi(a_t|s_t, s_{t+k}, k)$; x-axis represents the number of training iterations of dependency model. Four different colors represent four different actions. Blue line shows the KL divergence between predicted distribution and true distribution. (c): The blue line shows the mean KL divergence between $P^\pi(a_t|s_t, s_{t+k})$ and $P_\phi(a_t|s_t, s_{t+k}, k)$ over the dataset of random $(s_t, s_{t+k})$ pairs, averaged in 10 runs.

|  | Connected-low | Isolated-low | Connected-high | Isolated-high |
|---|---|---|---|---|
| MC | 0.63 | 1.66 | 6.21 | 16.58 |
| IS | 1.43 | 8.68 | 1.45 | 8.63 |
| IAE | 0.59 | 0.60 | 0.68 | 0.71 |

Table 1: Standard derivation of various estimators in different transition probability settings (connected and isolated) and different reward settings (low-variance and high-variance).

As illustrated in Figure 2a, the blue square represents the position of the agent and the yellow square represents the position of current goal. The agent gets positive reward for reaching the goal. To make the problem harder, the environment will do periodical reset multiple times in an episode, by which the agent and the goal are randomly repositioned. We use two different reward settings: per-step punishment setting and no punishment setting, which is detailed in Appendix C.1.

The hyper-parameters and network architectures we use in our experiments are presented in Appendix C.2.

## 8.2 DEPENDENCY FACTOR MODELING

In this section, we investigate our estimation of dependency factor $C^\pi$, and show the general similarity between our estimation and ground-truth $C^\pi$. We train our model $C_\phi$ by data generated by a fixed uniform random policy $\pi$. Figure 2a and 2b show the case where the dependency is precisely captured: given the future state $s_{t+k}$ shown in Figure 2a, the model $P_\phi(a_t|s_t, s_{t+k}, k)$ correctly predicts down and right action that more likely lead current state $s_t$ to future state $s_{t+k}$. We also build a dataset consisting of 300 random $(s_t, s_{t+k})$ pairs, where $k$ is uniformly sampled from 1 to 30. We evaluate the mean KL divergence between ground-truth $P^\pi(a_t|s_t, s_{t+k})$ and prediction $P_\phi(a_t|s_t, s_{t+k}, k)$ averaged in 10 runs, as shown in Figure 2c. The mean KL divergence decrease to a relatively small value during training, showing that the dependency model $P_\phi$ leads to a generally precise estimation of dependency factor $C^\pi$.

## 8.3 THE VARIANCE AND ACCURACY OF INDEPENDENCE-AWARE ADVANTAGE ESTIMATION

We evaluate the variance of IAE estimator on a variety of finite-state MDP settings. We train $r(s_t, s_{t+k})$ for 10000 episodes and then test the advantage estimator by performing advantage estimation multiple times to get the estimation variance. We compare the variance of IAE estimator with Monte-Carlo advantage estimator (MC) and importance sampling advantage estimator (IS). In this experiment, we use the precise value of dependency factor for IS and IAE estimator. Table 1

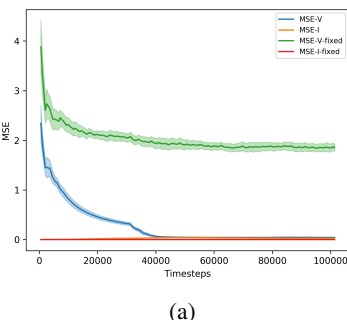 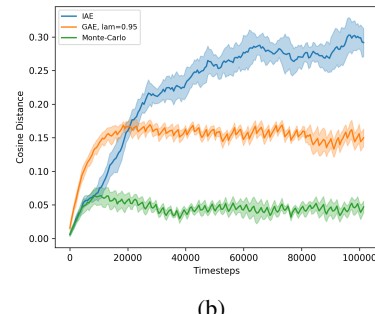

(a) (b)

Figure 3: (a): Mean squared error of two value functions $V_{w_1}$ and $I_{w_2}$ averaged in 10 runs on Pixel Grid World. Green and red lines show the MSE of $V_{w_1}$ and $I_{w_2}$ respectively, when $r_\psi$ is fixed; blue and orange lines show the MSE of $V_{w_1}$ and $I_{w_2}$ respectively, when $r_\psi$ is trained by our method. (b): The cosine similarity between advantage estimation and ground-truth advantage function. We compare IAE estimation, Monte-Carlo estimation and GAE estimation.

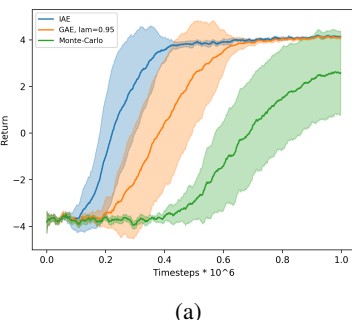 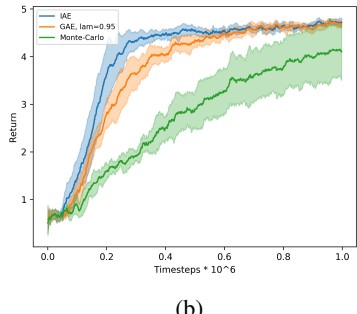

(a) (b)

Figure 4: Overall performance curve. Figure (a) and (b) respectively show the training curve on Pixel Grid World environment in per-step punishment setting and no punishment setting, averaged in 10 random seeds. In per-step punishment setting agent gets negative rewards before reaching goals, while in no punishment setting agent gets no reward before reaching goals.

demonstrates the standard derivation of advantage estimation. In both environments suitable for MC estimation and ones suitable for IS estimation, our method gives estimation with less variance than both MC and IS method. In some cases, IAE estimation dramatically reduces the variance of both MC and IS estimation.

On function approximation settings, we show that our method dramatically reduces the mean squared error in training value function approximators, as shown in Figure 3a. We initialize reward decomposition model $r_\psi$ to be zero for all inputs, which constructs a precise Monte-Carlo advantage estimator initially, and compare the value function training error with or without training $r_\psi$. When the reward decomposition model $r_\psi$ is fixed, the loss of value function training keeps to be high because of the stochasticity of Monte-Carlo return signal, while in our method, the $r_\psi$ makes reward to be distributed into importance sampling advantage estimator, making the mean squared error of value function to reduce. In Figure 3b, we show that IAE estimation has much higher cosine similarity to ground-truth advantage function, compared with Monte-Carlo and GAE estimation.

Besides, we investigate how independence property reduces the variance of estimation by visualizing the reward decomposition in Pixel Grid World Environment, which is detailed in Appendix C.3.

### 8.4 PERFORMANCE OF POLICY OPTIMIZATION

We run Proximal Policy Optimization algorithm with IAE advantage estimation method, and compare the result to PPO algorithm with Monte-Carlo and GAE advantage estimation. Figure 4 shows

the result on two different reward settings. Compared with Monte-Carlo and GAE estimation, IAE estimation makes the policy improvement process more sample-efficient.

## 9 CONCLUSIONS

In this work, we addressed the large variance problem in advantage estimation for policy gradient methods. We proposed a novel advantage estimation method by importance sampling, which identifies and utilizes the independence property, reducing the variance by ignoring those independent rewards. We further combined the proposed estimator with Monte-Carlo estimator in the optimal way, making the final IAE estimator to have low variance in general cases. The effectiveness of our method can be verified on Pixel-input environments compared with previous advantage estimation methods such as Monte-Carlo and Generalized Advantage Estimation.

There are a few directions to explore in the future. First, we only considered problems with discrete actions in this work, and we will extend our method to problems with continuous actions. Second, we will test our algorithm on more reinforcement learning environments, such as Atari games and continuous control tasks.

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

# A  PROOFS

## A.1  PROOF OF UNBIASEDNESS

In this section, we show the proof of theorem 1, which demonstrates that any choice of $r_{t+k}$ will not bias the combined estimator as long as $r_{t+k}$ is a function of $s_t$, $a_t$ and $\tau_{t+k}$.

**Theorem 1.** *Suppose $r_{t+k} \sim \hat{R}(r_{t+k}|s_t, a_t, \tau_{t+k})$, where $\hat{R}$ is any probability distribution. Then*

$$A^\pi(s_t, a_t) = \mathbb{E}_{\tau_t \sim P^\pi(\tau_t|s_t,a_t)} \left[ \sum_{k=0}^{T-t} \gamma^k (R_{t+k} - r_{t+k}) \right] - \mathbb{E}_{\tau_t \sim P^\pi(\tau_t|s_t)} \left[ \sum_{k=0}^{T-t} \gamma^k (R_{t+k} - r_{t+k}) \right]$$

$$+ \mathbb{E}_{\tau_t \sim P^\pi(\tau_t|s_t)} \left[ \sum_{k=0}^{T-t} \gamma^k r_{t+k} C_k^\pi(s_t, a_t, S_{t+k}, A_{t+k}) \right], \tag{1}$$

$$\text{where } C_k^\pi(s_t, a_t, s_{t+k}, a_{t+k}) := \frac{P^\pi(s_{t+k}, a_{t+k}|s_t, a_t)}{P^\pi(s_{t+k}, a_{t+k}|s_t)} - 1.$$

*Proof.* Denote $r(s_t, a_t, \tau_{t+k}) = \mathbb{E}[r_{t+k}|s_t, a_t, \tau_{t+k}]$.

Since $A^\pi(s_t, a_t)$ can be written by

$$A^\pi(s_t, a_t) = \mathbb{E}_{\tau_t \sim P^\pi(\tau_t|s_t,a_t)} \left[ \sum_{k=0}^{T-t} \gamma^k R_{t+k} \right] - \mathbb{E}_{\tau_t \sim P^\pi(\tau_t|s_t)} \left[ \sum_{k=0}^{T-t} \gamma^k R_{t+k} \right],$$

We only need to prove that

$$\mathbb{E}_{\tau_t \sim P^\pi(\tau_t|s_t)} \left[ \sum_{k=0}^{T-t} \gamma^k r_{t+k} C_k^\pi(s_t, a_t, S_{t+k}, A_{t+k}) \right]$$

$$= \mathbb{E}_{\tau_t \sim P^\pi(\tau_t|s_t,a_t)} \left[ \sum_{k=0}^{T-t} \gamma^k r_{t+k} \right] - \mathbb{E}_{\tau_t \sim P^\pi(\tau_t|s_t)} \left[ \sum_{k=0}^{T-t} \gamma^k r_{t+k} \right]. \tag{2}$$

We have derivation that

$$\mathbb{E}_{\tau_t \sim P^\pi(\tau_t|s_t)} \left[ \sum_{k=0}^{T-t} \gamma^k r_{t+k} C_k^\pi(s_t, a_t, S_{t+k}, A_{t+k}) \right] + \mathbb{E}_{\tau_t \sim P^\pi(\tau_t|s_t)} \left[ \sum_{k=0}^{T-t} \gamma^k r_{t+k} \right]$$

$$= \mathbb{E}_{\tau_t \sim P^\pi(\tau_t|s_t)} \left[ \sum_{k=0}^{T-t} \gamma^k r_{t+k} (1 + C_k^\pi(s_t, a_t, S_{t+k}, A_{t+k})) \right]$$

$$= \sum_{k=0}^{T-t} \int_S \sum_{a_{t+k}} \gamma^k \mathbb{E}_{\tau_{t+k} \sim P^\pi(\tau_{t+k}|s_{t+k}, a_{t+k})} \left[ r(s_t, a_t, \tau_{t+k}) \right] (1 + C_k^\pi(s_t, a_t, s_{t+k}, a_{t+k})) P^\pi(s_{t+k}, a_{t+k}|s_t) ds_{t+k}$$

$$= \sum_{k=0}^{T-t} \int_S \sum_{a_{t+k}} \gamma^k \mathbb{E}_{\tau_{t+k} \sim P^\pi(\tau_{t+k}|s_{t+k}, a_{t+k})} \left[ r(s_t, a_t, \tau_{t+k}) \right] \frac{P^\pi(s_{t+k}, a_{t+k}|s_t, a_t)}{P^\pi(s_{t+k}, a_{t+k}|s_t)} P^\pi(s_{t+k}, a_{t+k}|s_t) ds_{t+k}$$

$$= \sum_{k=0}^{T-t} \int_S \sum_{a_{t+k}} \gamma^k \mathbb{E}_{\tau_{t+k} \sim P^\pi(\tau_{t+k}|s_{t+k}, a_{t+k})} \left[ r(s_t, a_t, \tau_{t+k}) \right] P^\pi(s_{t+k}, a_{t+k}|s_t, a_t) ds_{t+k}$$

$$= \mathbb{E}_{\tau_t \sim P^\pi(\tau_t|s_t,a_t)} \left[ \sum_{k=0}^{T-t} \gamma^k r_{t+k} \right],$$

which proves equation 2 and further proves theorem 1. $\square$

## A.2 Upper Bound of Variance

In this section, we show that the if we use the sum of three random variable in equation 3 as the advantage estimator, the variance of advantage estimation is bounded by three times the sum of the variance of estimation individual random variable inside expectation in equation 3. We begin with the following lemma:

**Lemma 1.** *Suppose $X_1, X_2, \cdots, X_n$ are random variables. Then*

$$\text{Var}\left[\sum_{i=1}^{n} X_i\right] \leq n \sum_{i=1}^{n} \text{Var}[X_i].$$

*Proof.*

$$\text{Var}\left[\sum_{i=1}^{n} X_i\right] = \sum_{i=1}^{n} \text{Var}[X_i] + 2 \sum_{1 \leq i < j \leq n} \text{Cov}[X_i, X_j]$$

$$\leq \sum_{i=1}^{n} \text{Var}[X_i] + 2 \sum_{1 \leq i < j \leq n} \sqrt{\text{Var}[X_i]\text{Var}[X_j]}$$

Then we have

$$n \sum_{i=1}^{n} \text{Var}[X_i] - \text{Var}\left[\sum_{i=1}^{n} X_i\right]$$

$$\geq (n-1) \sum_{i=1}^{n} \text{Var}[X_i] - 2 \sum_{1 \leq i < j \leq n} \sqrt{\text{Var}[X_i]\text{Var}[X_j]}$$

$$= \sum_{1 \leq i < j \leq n} \left(\sqrt{\text{Var}[X_i]} - \sqrt{\text{Var}[X_j]}\right)^2 \geq 0,$$

which proves lemma 1. $\square$

From lemma 1, it is clear that the variance of advantage estimation is bounded by three times the sum of three individual variances shown by equation 5.

## A.3 Action probability form for Dependency Factor

In this section, we give detailed proof of equation 13 in the main body, which is written by:

$$\frac{P^\pi(s_{t+k}, a_{t+k}|s_t, a_t)}{P^\pi(s_{t+k}, a_{t+k}|s_t)} = \frac{P^\pi(a_t|s_t, s_{t+k}, a_{t+k})}{\pi(a_t|s_t)} = \frac{P^\pi(a_t|s_t, s_{t+k})}{\pi(a_t|s_t)}, \text{if } k \geq 1. \tag{3}$$

*Proof.* We separately prove the first and the second equation. The first equation is proved by:

$$\frac{P^\pi(s_{t+k}, a_{t+k}|s_t, a_t)}{P^\pi(s_{t+k}, a_{t+k}|s_t)} = \frac{P^\pi(s_{t+k}, a_{t+k}, s_t, a_t)}{P^\pi(s_t, a_t)} \cdot \frac{P^\pi(s_t)}{P^\pi(s_{t+k}, a_{t+k}, s_t)}$$

$$= \frac{P^\pi(s_{t+k}, a_{t+k}, s_t, a_t)}{P^\pi(s_{t+k}, a_{t+k}, s_t)} \cdot \frac{P^\pi(s_t)}{P^\pi(s_t, a_t)}$$

$$= \frac{P^\pi(a_t|s_t, s_{t+k}, a_{t+k})}{\pi(a_t|s_t)}.$$

The second equation is proved by:

$$
\begin{aligned}
P^\pi(a_t|s_t, s_{t+k}, a_{t+k}) &= \frac{P^\pi(s_t, a_t, s_{t+k}, a_{t+k})}{P^\pi(s_t, s_{t+k}, a_{t+k})} \\
&= \frac{P^\pi(s_t, a_t, s_{t+k})P^\pi(a_{t+k}|s_t, a_t, s_{t+k})}{P^\pi(s_t, s_{t+k})P^\pi(a_{t+k}|s_t, s_{t+k})} \\
&= \frac{P^\pi(s_t, a_t, s_{t+k})\pi(a_{t+k}|s_{t+k})}{P^\pi(s_t, s_{t+k})\pi(a_{t+k}|s_{t+k})} \\
&= \frac{P^\pi(s_t, a_t, s_{t+k})}{P^\pi(s_t, s_{t+k})} = P^\pi(a_t|s_t, s_{t+k}).
\end{aligned}
$$

$\square$

Thus, it is proved that $a_{t+k}$ can be abandoned from the probability condition when $k \geq 1$. When $k = 0$, the conditional probability can be directly given by a closed form solution:

$$
P^\pi(A_t = a|s_t, s_{t+k}, a_{t+k}) = \begin{cases} 1, & a = a_{t+k} \\ 0, & a \neq a_{t+k} \end{cases}
$$

### A.4 DEPENDENCY FACTOR ESTIMATION

We prove that the dependency factor can be learned in an inverse temporal difference manner in this section, which is formulated by equation 14 in the main body:

$$
P^\pi(a_t|s_t, s_{t+k_2}) = \mathbb{E}_{s_{t+k_1} \sim P^\pi(s_{t+k_1}|s_{t+k_2}, s_t)} P^\pi(a_t|s_t, s_{t+k_1}), \text{if } k_2 > k_1 \geq 1.
$$

*Proof.*

$$
\begin{aligned}
&\mathbb{E}_{s_{t+k_1} \sim P^\pi(s_{t+k_1}|s_{t+k_2}, s_t)} P^\pi(a_t|s_t, s_{t+k_1}) \\
&= \int_S P^\pi(s_{t+k_1}|s_{t+k_2}, s_t) P^\pi(a_t|s_t, s_{t+k_1}) ds_{t+k_1} \\
&= \int_S \frac{P^\pi(s_{t+k_1}, s_{t+k_2}, s_t)}{P^\pi(s_{t+k_2}, s_t)} \frac{P^\pi(a_t, s_t, s_{t+k_1})}{P^\pi(s_t, s_{t+k_1})} ds_{t+k_1} \\
&= \int_S \frac{P^\pi(a_t, s_t, s_{t+k_1})}{P^\pi(s_{t+k_2}, s_t)} P^\pi(s_{t+k_2}|s_{t+k_1}, s_t) ds_{t+k_1} \\
&= \int_S \frac{P^\pi(a_t, s_t, s_{t+k_1})}{P^\pi(s_{t+k_2}, s_t)} P^\pi(s_{t+k_2}|s_{t+k_1}) ds_{t+k_1} \\
&= \int_S \frac{P^\pi(a_t, s_t, s_{t+k_1}, s_{t+k_2})}{P^\pi(s_{t+k_2}, s_t)} ds_{t+k_1} \\
&= \frac{P^\pi(a_t, s_t, s_{t+k_2})}{P^\pi(s_{t+k_2}, s_t)} = P^\pi(a_t|s_t, s_{t+k_2})
\end{aligned}
$$

$\square$

## B TEMPORAL DIFFERENCE LEARNING OF DEPENDENCY MODEL

We use a mixture of temporal difference and Monte-Carlo training target to train dependency model. For a trajectory sample $\tau = \{s_i, a_i, r_i\}_{i=1}^T$, we provide training target for $P_\phi(a_t|s_t, s_{t+k}, k)$ for every valid $(t, k)$ pair, and use cross-entropy loss to update $\phi$. For the Monte-carlo training target, we directly use one-hot probability distribution of action $a_t$ as the ground-truth; for the temporal difference training target, we use distribution of $P_\phi(a_t|s_t, s_{t+k-1}, k - 1)$ as the ground-truth. We mix the two training target with $0.85^k$ weight on Monte-Carlo target and $1 - 0.85^k$ weight on temporal difference target to achieve the optimal bias-variance tradeoff.

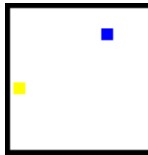

Figure 5: Observation in Pixel Grid World.

## C EXPERIMENT

### C.1 ENVIRONMENT SETTINGS

**Finite-state MDP settings.** We construct different 3-state MDPs with different transition probability and reward functions. We categorize transition probability settings into two types:

- Connected settings, where there is high mutual reaching probability between every state pair, causing the n-step transition probability to converge rapidly into stationary distribution. We use the following transition matrix for connected settings:

$$P = \begin{bmatrix} 0.2 & 0.5 & 0.3 \\ 0.5 & 0.3 & 0.2 \\ 0.2 & 0.2 & 0.6 \end{bmatrix}$$

- Isolated settings, where there is low mutual reaching probability between some state pairs, causing the n-step transition probability to converge slowly into stationary distribution. We use the following transition matrix for isolated settings:

$$P = \begin{bmatrix} 0.9 & 0.05 & 0.05 \\ 0.45 & 0.1 & 0.45 \\ 0.05 & 0.05 & 0.9 \end{bmatrix}$$

In connected settings, the n-step transition probability is very close into stationary distribution, resulting in a small dependency factor in future rewards. Thus, importance sampling estimator will have lower variance in connected settings than in isolated settings.

We also categorize reward settings into two types:

- High-variance settings, where the variance of Monte-Carlo return signal is large compared to mean total return. We use the following reward function for high variance settings:

$$R = \begin{bmatrix} 3 & 2 & 1 \end{bmatrix}$$

- Low-variance settings, where the variance of Monte-Carlo return signal is small compared to mean total return. We use the following reward function for low variance settings:

$$R = \begin{bmatrix} 2.1 & 2 & 1.9 \end{bmatrix}$$

It is clear that Monte-Carlo advantage estimator will have larger variance in high-variance settings than in low-variance settings.

**Pixel Grid World environment.** We build Pixel Grid World environment where observations are provided by high-dimensional images. As shown in Figure 5, the blue square represents the position of the agent and the yellow square represents the position of current goal. The size of grid world is $8 \times 8$, and observations are provided by $128 \times 128 \times 3$ RGB pixels. There are 4 actions in Pixel Grid World, making the agent to move toward 4 different directions. We use two reward settings in experiments:

- Per-step punishment setting, where agent gets $r = -0.03$ reward in every step before reaching goal, and $r = 0$ reward in every step after reaching goal. Agent gets $r = 1$ reward when reaching goal for the first time.
- No punishment setting, where agent gets $r = 1$ reward when reaching goal for the first time, and gets $r = 0$ reward otherwise.

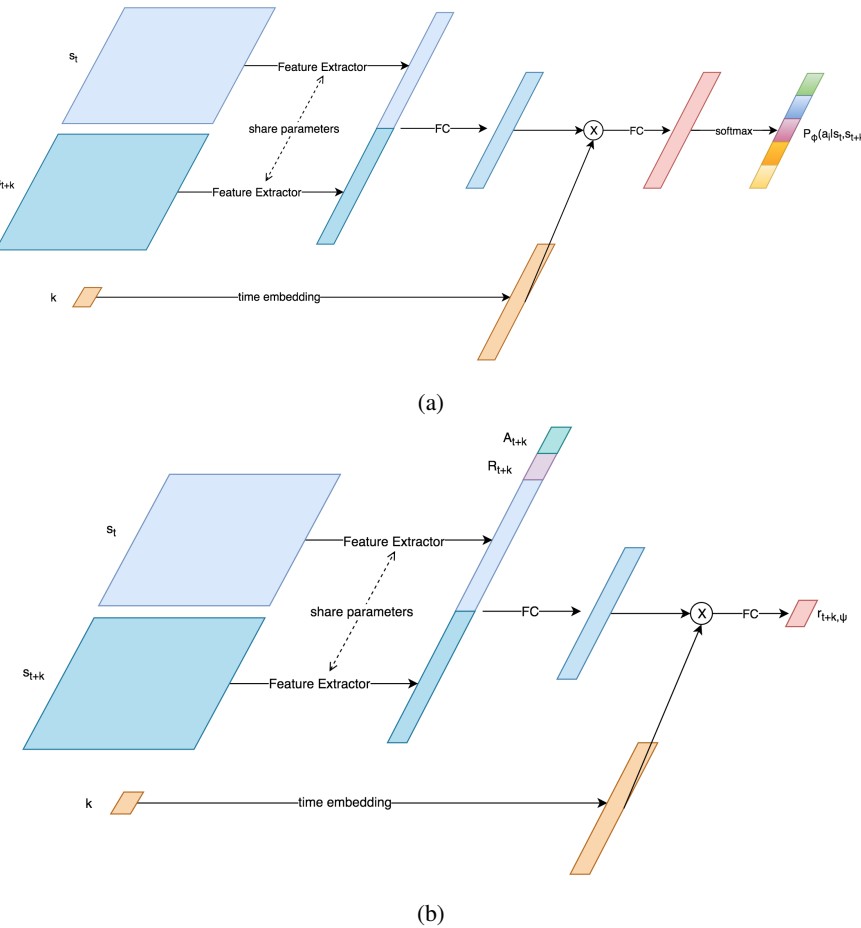

Figure 6: Network architecture for dependency model and reward decomposition model.

To make the problem harder, the environment will do periodical reset multiple times in an episode, by which the agent and the goal are randomly repositioned. The reset period is set to be 30 steps, and an episode contains 5 rounds.

## C.2 HYPER-PARAMETER SETTINGS AND NETWORK ARCHITECTURE

We use $\gamma = 0.999$ for all experiments in policy optimization. For GAE estimator, we use the default settings $\lambda = 0.95$ in all experiments. We use 8 parallel environments to gather data, use the trajectory truncation length of 128, and train each truncated trajectory with 5 epochs. We use a global Adam optimizer for training both dependency model and reward decomposition model, and set learning rate to be $2.5 \times 10^{-4}$. For initialization, we use orthogonal initialization with $\sqrt{2}$ scale for all layers, except that we initialize the last weight matrix of D to be zeros. For training $P_\phi(a_t|s_t, s_{t+k}, k)$, we weight the MC loss by $0.85^k$ and TD loss by $1 - 0.85^k$, and use the mixture loss to train dependency model. For PPO algorithm, we use the same hyper-parameter settings with OpenAI Baseline.

We use the network architecture shown in Figure 6 for dependency model and reward decomposition model. We use the same CNN feature extractor settings as the settings in the OpenAI Baseline. For time embedding, we store embedding vector of 128 dimensions for each $k \in \{1, 2, \cdots, 128\}$ separately.

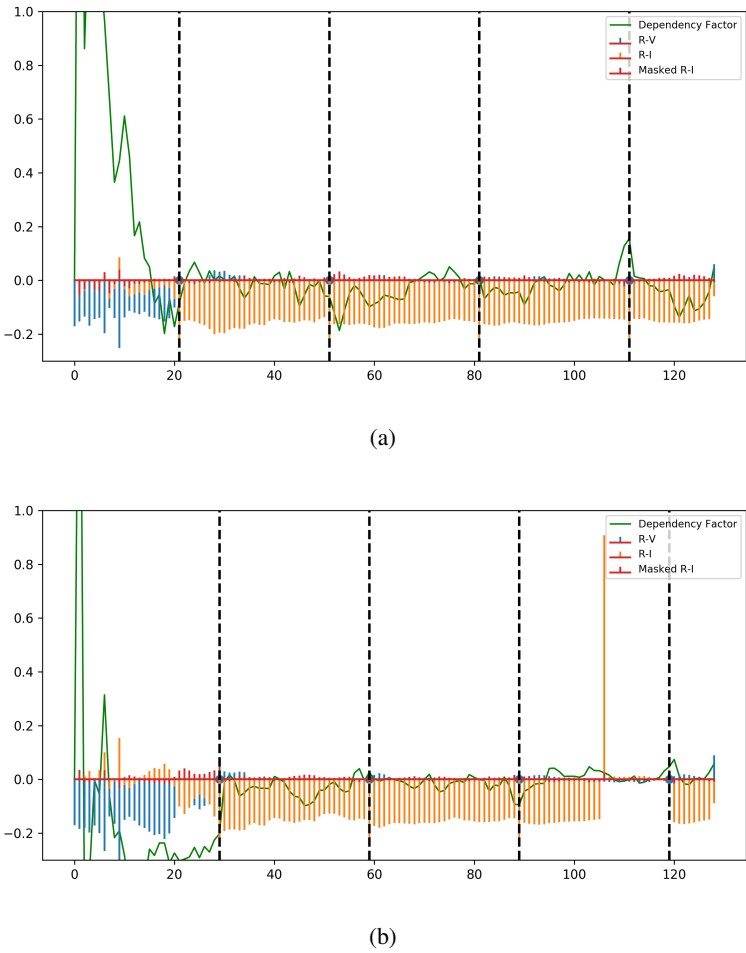

(a)

(b)

Figure 7: Case study on reward decomposition model. We fix a state-action pair $(s_t, a_t)$ and visualize reward decomposition for every future reward. The x-axis represents the interval from current state $s_t$ to future states $s_{t+k}$, green line represents the dependency factor of future states, blue and orange lines are future rewards that assigned to Monte-Carlo estimator and importance sampling estimator respectively, and red lines are the estimation by importance sampling estimator on each future time. Black dashed lines represent periodical resets.

## C.3 CASE STUDY ON REWARD DECOMPOSITION MODEL

In Pixel Grid World environment, we fix a stochastic policy and train model $r_\psi$ to minimize estimation variance. When the model $r_\psi$ is near convergence, we visualize reward decomposition for every future reward after a state-action pair $(s_t, a_t)$. These results are shown in Figure 7. It is clear that all the independent rewards after reset are assigned to importance sampling estimator, which is multiplied by a small dependency factor and barely contribute to advantage estimation. By contrast, rewards before reset are assigned to the Monte-Carlo estimator, which contribute to advantage estimation.

