# OpenReview forum: "Independence-aware Advantage Estimation"
_ICLR.cc/2020/Conference — Reject_

### Official Review · AnonReviewer2 · 2019-10-22
**Official Blind Review #2**

**Rating:** 6

**Review:**

This paper proposes a novel advantage estimate for reinforcement learning based on estimating the extent to which past actions impact the current state. More precisely, the authors train a classifier to predict the action taken k-steps ago from the state at time t, the state at t-k and the time gap k. The idea is that when it is not possible to accurately predict the action, the action choice had no impact on the current state, and thus should not be assigned credit for the current reward, they refer to this as the "independence property" between current action and future states. Based on this idea, the authors introduce a "dependency factor", using the ratio P(s_{t+k},a_{t+k}|s_t,a_t)/P(s_{t+k},a_{t+k}|s_t). They later show that this can be reworked using Bayes theorem into a ratio of the form P(a_t|s_t,s_{t+k})/\pi(a_t|s_t) which is more convenient to estimate. The authors show mathematically that, when the dependency factor is computed with the true probabilities and use to weight each reward in a trajectory, the result is an unbiased advantage estimator. Importantly the expectation, in this case, is taken over trajectories sampled according to the policy pi conditioned only on S_t. This is distinct from the Monte-Carlo estimator which is based only on samples in which A_t, the action whose advantage is being estimated, was selected.

They go on to say that this estimator will tend to have lower variance than the conventional Monte-Carlo estimator when future rewards are independent of current actions. However, the variance can actually be higher, due to the importance sampling ratio used, when future rewards are highly dependent on the current action. They propose a method to combine the two estimators on a per reward while maintaining unbiasedness using a control variate style decomposition. This introduces a tunable reward decomposition parameter which determines how to allocate each reward between the two estimators. The authors propose a method to tune this parameter by approximately optimizing an upper bound on the variance of the combined estimator.

As a final contribution, the authors introduce a temporal-difference method of estimating the action probability P(a_t|s_t,s_{t+k}) required by their method.

In the experiments, the authors provide empirical evidence that various aspects of their proposed method can work as suggested on simple problems. They also provide a simple demonstration where their advantage estimator is shown to improve sample efficiency in a control problem.

This paper suffers from moderate clarity issues, but I lean toward acceptance primarily because I feel that the central idea is a solid contribution. The idea of improving credit assignment by explicitly estimating how much actions impact future states seems reasonable and interesting. The temporal difference method introduced for estimating P(a_t|s_t,s_{t+k}) is also interesting and clever. I'm less confident in the introduced method for trading off between the Monte Carlo and importance sampling estimators. The experiments seem reasonably well executed and do a fair job of highlighting different aspects of the proposed method.

The derivation of the combined estimator was very confusing to me. It's strange that the derivation of the variance lower bound includes terms which are drawn from both a state conditional and state-action conditional trajectory. You're effectively summing variances computed with respect to two different measures, but the quantity being bounded is referred to as just the "variance of the advantage estimator". What measure is this variance supposed to be computed with respect to? Especially given that as written the two estimators rely on samples drawn from two different measures. It doesn't help that the advantage estimator whose variance is being constructed is never explicitly defined but just referred to as "advantage estimator derived from equation 3". Nevertheless, if we ignore the details of what exactly it is a lower bound of, the sum of the three variances in equation 5 seems a reasonable surrogate to minimize.

Related to the above point I don't fully understand what the variances shown in table 1 mean in the experiments section. For the IAE estimator for example, is the variance computed based on each sample using three independent trajectories (one for each term) or is it computed from single trajectories? If it's from single trajectories I can't understand how the expression would be computed.

Questions for the authors:
-Could you please explicitly define the "advantage estimator derived from equation 3"?
-Could you please explain precisely how the variance is computed in table 1?

Update:

Having read the other reviews and authors response I will maintain my score of a weak accept, though given more granularity I would raise my score to a 7 after the clarifications. I appreciate the authors' clarification of the advantage estimator and feel the related changes to the paper are very helpful. I still feel the central idea of the work is quite strong.

However, I also feel the control variate part of the work is very loosely specified. In particular, given the use of function approximation in practice instead of actually sampling 3 trajectories the validity of the control variate method applied is questionable. As the authors say "if the random variable in either term has high variance, function approximators will tend to have large error", This may be true initially but the function approximator can already reduce variance over time by learning, so it's not clear how the function approximators and control variate complement each other. This is something I feel would be worthwhile to explore more in future work.

Also, I feel it's worth pointing out that a concurrent paper presenting a very similar idea is scheduled to be presented at NeurIPS 2020, which can be found here: https://papers.nips.cc/paper/9413-hindsight-credit-assignment. I don't feel this in any way undermines the contribution of the work presented here, but merely wanted to make the meta reviewer aware in case it was relevant to their decision. In fact, I feel this work complements that one in a number of ways, including the presentation of the temporal difference method for learning the action probabilities.

**Experience Assessment:**

I have published one or two papers in this area.

**Review Assessment: Checking Correctness Of Derivations And Theory:**

I carefully checked the derivations and theory.

**Review Assessment: Checking Correctness Of Experiments:**

I assessed the sensibility of the experiments.

**Review Assessment: Thoroughness In Paper Reading:**

I read the paper thoroughly.

---

> ### Author Response · Authors · 2019-11-09
> **Response to Reviewer 2**
>
> Thank you so much for your supportive comments. In the following sections, we will give response to two questions you have. Please let us know if you have any questions in the response.
>
> [Define the "advantage estimator derived from equation 3”]
>
> We are sorry that we didn't clearly define the advantage estimator without function approximations. In our updated version of paper, the advantage estimator is defined formally, based on one assumption that we can sample multiple trajectories on the same state $s_t$. If three samples are individually sampled in every expectation term in equation (3), we are able to define an advantage estimator with the variance in the same form as equation (5) in the old version of paper, except that the factor is 1. The rest of discussions, which focuses on how to minimize the variance, will not be affected by this change.
>
> If we consider the problem practically, since we cannot sample multiple trajectories from the same state, we must use function approximators to represent some of values in equation (3). That doesn't mean it is unnecessary to consider the variance in either of three terms: if the random variable in either term has high variance, function approximators will tend to have large error. Thus practically, we think that using the sum of three variances as a surrogate objective will be a reasonable choice.
>
> [How variance is computed in table 1]
>
> We are sorry that we didn’t clearly mention this. For all three methods, we estimate the advantage function of the same state-action pair $(s, a)$. For IAE, we individually sample three trajectories for each estimation of $A(s, a)$, one starting from $(s, a)$ and two starting from s, and use these three trajectories to compute the three terms in equation (3). We have added these details to the experiment section in the updated version of our paper.

---

> > ### Comment · AnonReviewer2 · 2019-11-15
> > **Thank You**
> >
> > Thank you for your response, I'm satisfied that I now have a good understanding of the methods and experiments.

---

### Official Review · AnonReviewer1 · 2019-10-23
**Official Blind Review #1**

**Rating:** 6

**Review:**

This paper proposes a new advantage estimator in reinforcement learning based on importance sampling. This form allows for a significantly lower-variance estimator for situations where the current action "stops mattering" to the future state. A control variate, as in Grathwohl et al., is used to combine the importance sampling estimator with the "standard" estimator in a way that is always unbiased and attempts to minimize the overall variance.

The overall setting makes sense. I found your example (in the second paragraph of the introduction) initially somewhat misleading, though: in the setting where a game is composed of fully independent rounds, surely these would simply be modeled as completely separate MDPs. Even if not, settings where the rounds are reset after a variable length of time (e.g. the round ends when one player achieves some objective) would *not* fit the exact independence structure you assume at the start of Section 3, if your current action affects when the game will reset. But of course your estimator does not rely on actual *independence* (C = 0); it can take advantage of only "weak dependence" (and moreover this dependence need not be pre-specified). You might think about emphasizing this a little more in the introduction to emphasize that the estimator is general, and you're looking for one that can take advantage of these kinds of situations.

It might be worth noting after (2) that $C^\pi_k$ is upper-bounded by $1 / P^\pi(a_t \mid s_t) - 1$, so that the importance sampler is always well-defined and unbiased when action probabilities are nonzero. This does raise an issue: a policy which *ever* deterministically avoids an action, i.e. $\pi(a_t \mid s_t)$ in (13) is 0, will break the method. This is worth explicitly stating somewhere.

Something worth thinking about a bit: any choice of weights for your control variate provably doesn't affect your estimator in expectation (and you try to decrease its variance), so that bad estimation of e.g. the quantities in (7) won't lead you to being "incorrect," just higher-variance. But a bad choice of parameters in your $C^\pi$ estimator *would* bias your estimates. This is in some ways the same as the effect of using a value function or $Q$-function approximator, but can we say anything about the ways in which a bad $C$ estimator would likely affect the overall optimization process, perhaps in some very simple case? Would an unbiased $C$ estimator lead to an unbiased advantage estimator? (Not that it's clear how to get an unbiased estimator of the ratio in $C$ anyway.)

Some minor points on notation: Using $r_{t+k}$ for the control variate was initially confusing to me, because elsewhere you've used e.g. $S_t$ for the random variable of a state and $s_t$ as the value of that state -- it made me think that $r_{t+k}$ was somehow supposed to be the value of a reward $R_{t+k}$. Another letter might be better. Similarly, $V_{w_1}(s_t)$ of (7) isn't really a value function; it's the difference between the value function and the sum of discounted control variates. Also, $C_\phi$ doesn't estimate $C^\pi$: it estimates $C^\pi + 1$, so it might make more sense notationally to just subtract one from the definition of $C_\phi$.

Overall: I think the idea in this paper is sensible, the derivations fairly clear, and it seems to help empirically. It does add a lot of "moving parts" to the already-complicated RL setup, though, and I'm not well-versed enough in the RL literature to have much of a sense of how convincing these experiments are; hopefully another reviewer is.

**Experience Assessment:**

I do not know much about this area.

**Review Assessment: Checking Correctness Of Derivations And Theory:**

I assessed the sensibility of the derivations and theory.

**Review Assessment: Checking Correctness Of Experiments:**

I assessed the sensibility of the experiments.

**Review Assessment: Thoroughness In Paper Reading:**

I read the paper at least twice and used my best judgement in assessing the paper.

---

> ### Author Response · Authors · 2019-11-08
> **Response to Reviewer 1**
>
> Thank you so much for your supportive comments. In the following sections, we will give response to every question you have. Please let us know if you have any questions in the response.
>
> [Parts of the example in the introduction is misleading]
>
> Firstly, thanks for your advice on the example in the introduction. We have clarified that the time of each round is constant in the updated version of our paper. Also, it is worth mentioning that when the rounds last a variable length of time, there’s essentially no way to use independence property unless $\gamma=1$. That’s because the next round will happen in different time intervals, which will make the next round necessary to be considered. When $\gamma=1$, there will be ways to utilize independence property when you have variable length of time, but it can be left as future works.
>
> [The cases where the IS weight is not well defined]
>
> Thanks for the advice and we may add the discussion on the validity of importance sampling, and the bound of importance sampling weight in the camera-ready version. It is true that when $\pi(a_t|s_t)=0$, the IS-based advantage estimator on $(s_t, a_t)$ is not properly defined. However, we will never sample these actions and we will never need to estimate the advantage function on these state-action pairs in policy optimization process, so it won’t affect the practical performance.
>
> [How a bad C estimation affects optimization]
>
> It is true that a bad estimation in dependency factor will affect the policy optimization process in some cases; however, we will explain that not all bias in dependency factor estimation will harm the overall optimization process. For instance, we initialize the classifier $P_\phi(a_t|s_t, s_{t+k})$ to have the same output with policy $\pi(a_t|s_t)$. When there are not enough data to train the classifier, we will have all dependency factor to be close to zero, and all estimated advantage function becomes zero. Although certainly there are biases in advantage estimation in this case, it won't hurt the overall optimization process since the policy will remain unchanged. As there are enough data to train the classifier, the classifier $P(a_t|s_t, s_{t+k})$ can utilize the data to make more precise estimation, and the bias of advantage estimation will slowly approach to zero as estimation on C get more precise.
>
> [Will an unbiased C estimator lead to an unbiased advantage estimator]
>
> In our method, unbiased advantage estimation is based on not only accurate estimation on C, but also the assumption that we are able to draw two samples from the same state $s_t$: one sample from $P^\pi(\tau_t|s_t)$ and one sample from $P^\pi(\tau_t|s_t,a_t)$. Empirically, we have to use function approximations to avoid drawing multiple samples from the same state, then there’re inevitable biases in practice.
>
> [Confusions in notations]
>
> We agree that these notations are not very clear, and highly appreciate the comments. We have modified these notations in the updated version of our paper.

---

> > ### Comment · AnonReviewer1 · 2019-11-15
> > **Thanks**
> >
> > Thanks for your comments. I'm satisfied, and also think the improvements from R3's review were helpful.

---

### Official Review · AnonReviewer3 · 2019-10-25
**Official Blind Review #3**

**Rating:** 3

**Review:**

This paper tries to control the variance of advantage function by utilizing the independence property between current action and future states. The practical approach they are using is to learn a dependency model of reward function as a control variate to lower the variance. Using the control variate technique they derive a (maybe complicated) algorithm to update the policy by PPO. Empirical results seems competitive.

The major concern is the novelty. It is similar to many of the control variate technique papers (e.g. Liu et al. (2017)), which learn a model to decrease the variance in a certain way. I don't see from the paper for the advantage of applying control variate over advantage function compared to previous methods.

The minor concern is for the clarity. Section 4 is unsatisfactory. The derivation seems correct to me, however, there are too many parameters introduce to optimize, which I cannot directly get the main page of what the algorithm is doing for the first time read. From the algorithm box, there are totally 5 different parameter: $\theta, \phi, w_1, w_2$ and $\psi$ to update which make the algorithm pretty messy. I believe there are better way to either get a simpler algorithm or demonstrate your algorithm in a better way. Section 5 is even harder to understand. Could you explain why equation (14) is similar to Liu ei al. 2018?

Overall I tends to reject the paper at the moment. I encourage the authors to do more surveys on control variate technique in policy optimization and highlight the novelty of why controlling the variance of the advantage function can help to boost the performance of policy optimization.

Reference:
1. Liu, Hao, et al. "Action-depedent Control Variates for Policy Optimization via Stein's Identity." arXiv preprint arXiv:1710.11198 (2017).

**Experience Assessment:**

I have read many papers in this area.

**Review Assessment: Checking Correctness Of Derivations And Theory:**

I carefully checked the derivations and theory.

**Review Assessment: Checking Correctness Of Experiments:**

I assessed the sensibility of the experiments.

**Review Assessment: Thoroughness In Paper Reading:**

I read the paper at least twice and used my best judgement in assessing the paper.

---

> ### Author Response · Authors · 2019-11-08
> **Response to Reviewer 3**
>
> Thank you so much for your detailed comments. In the following parts, we will give the responses for each concern you have. Please let us know if you have any questions in the response.
>
> [The concern of novelty]
>
> We think the control variate technique is not the novel part in our works. Instead, we think that the novelty of our work mainly lies on the novel estimator being proposed, and the independence property it utilizes to reduce variance. When independence property exists, i.e. $P(s_{t+k}|s_t) = P(s_{t+k}|s_t,a_t)$, our method can utilize the independence property to reduce the variance of advantage estimation, which has never been proposed by previous work to the best of our knowledge. The control variate technique is only used to augment the novel important sampling estimator by combining it with the MC estimator.
>
> There are significant differences in the estimator being used, between our method and previous works of variance reduction methods in policy gradients [1,2]. The first paper (Liu et al. (2017)), decomposing the REINFORCE policy gradient estimator partly into the deterministic policy gradient estimator, relies on the setting that the action space is continuous. Because the difference lies in the basic parts, there can be great difference in most of aspects: the method of Liu et al. cannot address the high-variance problem of Monte-Carlo return estimator, which still appears in the final form (in equation (9) of the paper), however our method can avoid the high variance problem of Monte-Carlo estimator when independence property exists; the method of Liu et al. don’t have an extension on discrete actions space, however our method is designed for discrete action space. The second paper (Wu et al. (2018)) relies on another property that each dimension of the continuous action are individually sampled from the policy, which gives a new form of baseline function (baseline is also a form of control variate), which also results in a method essentially different than ours.
>
> Although we think there’re a lot of differences between our method and previous works, we highly agree with your comments that these previous works share a common part with our works, such as the usage of control variate. We have added discussion in the related work section about variance reduction methods in policy gradient [1,2], and their relationship and difference with our methods.
>
> [The concern of clarity]
>
> We are sorry that section 4 can be hard to read. In section 4, we essentially make two definitions of random variable: $\hat{J}$ and $\hat{I}$, and also define two function approximators: $V_{w_1}$ and $I_{w_2}$ which approximates the expectation of two random variable. After these definitions, the rest of thing we do in section 4 is to find the gradient to optimize reward decomposition. In future version of our paper, we will make these definitions clearer and separate all the related discussions of optimizing reward decomposition into one subsection, to make it easier to understand.
>
> [The interpretation of the temporal difference updates for the dependency model]
>
> Consider the theorem 1 in the paper of Liu et al, if you set $\pi$ and $\pi_0$ to be the same policy except $\pi$ starting from $(s, a)$ and $\pi_0$ starting from $s$, you will have $\beta(a|s)=1$ and $w(s)$ being the analogy of dependency factor, although considered on stationary distributions. The analogy of dependency factor in s’ is the expected value of dependency factor in $s$, when $s$ is sampled from the time-reversed conditional probability. Since the settings are different, it is better to say that we are inspired by the idea, and our temporal difference property can actually be considered a new proof.
>
> There are also intuitive ways to interpret equation (14): large $P(a_t|s_t,s_{t+k2})$ means $a_t$ causes $s_{t+k2}$ to happen; large $P(a_t|s_t,s_{t+k1})$ means $a_t$ causes $s_{t+k1}$ to happen; if all the $s_{t+k1}$ that causes $s_{t+k2}$ satisfies that $s_{t+k1}$ is caused by $a_{t}$, then $s_{t+k2}$ is caused by $a_{t}$.
>
> [1] Liu, Hao, et al. "Action-depedent Control Variates for Policy Optimization via Stein's Identity." arXiv preprint arXiv:1710.11198 (2017).
> [2] Wu, Cathy, et al. "Variance reduction for policy gradient with action-dependent factorized baselines." arXiv preprint arXiv:1803.07246 (2018).

---

> > ### Comment · AnonReviewer3 · 2019-11-14
> > **Thank you for the response**
> >
> > Thank you for the detailed response, it answer most of my concerns. Here are a few comment and follow up question I would like you to further address.
> >
> > > The concern of novelty.
> >
> > I like the intuition when $P(s_{t+k}|s_t) = P(s_{t+k}|s_t, a_t)$ we can reduce variance, that is a very clever observation! And I agree the part that two control variates methods we mentioned are quite different from yours. But I'm still quite confused on equation (1) and (3). Is equation (1) already a new estimator?
> >
> > I believe the intention for equation (3) is to combine importance sampling estimator and MC estimator, which is quite similar to doubly robust estimator (https://arxiv.org/pdf/1511.03722.pdf) except that you estimate the value for every (s,a) pair and used a learned model for the importance ratio. Could you explain their connection a little bit?
> >
> > Another question is that the estimator in equation (3) is not unbiased once you use TD to approximate $C^\pi$. This may bring additional question on bias variance tradeoff, which may need more theoretical analyses.
> >
> > > The concern of clarity
> > I have to admit that it is not easy for me to understand section 4 at the first few scans. I encourage the author to submit a revision version on that section before rebuttal section deadline. You should highlight what is the objective you want to optimize before writing a bunch of updating equation. It is hard to follow why the gradient should look like in this way without knowing the objective.

---

> > > ### Author Response · Authors · 2019-11-14
> > > **Thank you for your reply**
> > >
> > > Thank you for your reply! In the following section we will give response to all your concerns.
> > >
> > > [About the novelty of equation (1)]
> > > The equation (1) is already a new derivation, and we make these derivations independently. The property it uses to reduce variance, i.e. independence property, is also not been proposed as far as we know.
> > >
> > > [About doubly robust estimation]
> > > We agree that doubly robust estimation is also an effective way to combine two estimators, and it also uses control variate techniques which is similar to our method. In the following parts I will briefly talk about the difference. Their method is essentially combining two estimators: one biased estimator with low variance (i.e. $\hat{Q}$ in equation (10) of the paper) and one unbiased estimator with high variance (i.e. IS estimator in equation (5) of the paper), to get an estimator with low variance and no bias. Also, both IS estimator and $\hat{Q}$ estimator have the fixed form, with fixed variance and fixed bias. The case in our method is different. We decompose $A(s_t, a_t)$ into two estimators given any decomposition parameter $\psi$, where the variance of each estimator is determined by $\psi$. Then, the two estimators are actually determined by the decomposition parameter; and we want to tune decomposition parameter to make both of the estimator having low variance. From these perspectives, we think that our method differs from doubly robust estimators from the motivation.
> > >
> > > [About the bias caused by inaccuracy in dependency factor]
> > > We really agree that how bias will affect advantage estimation need further analysis, and we will consider giving a analysis in the aspect of mean squared error, which consider both bias and variance. Also, we have made an intuitive analysis on how a biased dependency factor estimation can affect policy optimization, which is detailed in the next paragraph. We hope these analysis can help you with your question.
> > >
> > > "It is true that a bad estimation in dependency factor will affect the policy optimization process in some cases; however, we will explain that not all bias in dependency factor estimation will harm the overall optimization process. For instance, we initialize the classifier $P_\phi(a_t|s_t,s_{t+k})$ to have the same output with policy $\pi(a_t|s_t)$. When there are not enough data to train the classifier, we will have all dependency factor to be close to zero, and all estimated advantage function becomes zero. Although certainly there are biases in advantage estimation in this case, it won't hurt the overall optimization process since the policy will remain unchanged. As there are enough data to train the classifier, the classifier $P(a_t|s_t,s_{t+k})$ can utilize the data to make more precise estimation, and the bias of advantage estimation will slowly approach to zero as estimation on C get more precise. "
> > >
> > > [About clarity of section 4]
> > > We have uploaded an updated version of our paper, where we separate all the content into three subsections. We also highlight the form of our objective function to optimize $\psi$ before writing the updating equation. We hope that the updated version can be clearer for you.
> > >
> > > We hope our response and the revision could address your concerns. Please let us know if you have any question on our response.

---

### Decision · Program_Chairs · 2019-12-19

**Decision:**

Reject

**Comment:**

Policy gradient methods typically suffer from high variance in the advantage function estimator. The authors point out independence property between the current action and future states which implies that certain terms from the advantage estimator can be omitted when this property holds. Based on this fact, they construct a novel important sampling based advantage estimator. They evaluate their approach on simple discrete action environments and demonstrate reduced variance and improved performance.

Reviewers were generally concerned about the clarity of the technical exposition and the positioning of this work with respect to other estimators of the advantage function which use control variates. The authors clarified differences between their approach and previous approaches using control variance and clarified many of the technical questions that reviewers asked about.

I am not convinced by the merits of this approach. While, I think the fundamental idea is interesting, the experiments are limited to simple discrete environments and no comparison is made to other control variate based approaches for reducing variance. Furthermore, due to the function approximation which introduces bias, the method should be compared to actor critic methods which directly estimate the advantage function. Finally, one of the advantages of on policy policy gradient methods is its simplicity. This method introduces many additional steps and parameters to be learned. The authors would need to demonstrate large improvements in sample efficiency on more complex tasks to justify this added complexity. At this time, I do not recommend this paper for acceptance.